# Synthesis and Biological Evaluation of Thiazole-Based Derivatives with Potential against Breast Cancer and Antimicrobial Agents

**DOI:** 10.3390/ijms23179844

**Published:** 2022-08-30

**Authors:** Ekaterina Pivovarova, Alina Climova, Marcin Świątkowski, Marek Staszewski, Krzysztof Walczyński, Marek Dzięgielewski, Marta Bauer, Wojciech Kamysz, Anna Krześlak, Paweł Jóźwiak, Agnieszka Czylkowska

**Affiliations:** 1Institute of General and Ecological Chemistry, Lodz University of Technology, Żeromskiego 116, 90-924 Łódź, Poland; 2Department of Synthesis and Technology of Drugs, Medical University, Muszyńskiego Street 1, 90-145 Łódź, Poland; 3Department of Inorganic Chemistry, Faculty of Pharmacy, Medical University of Gdańsk, 80-416 Gdańsk, Poland; 4Department of Cytobiochemistry, Faculty of Biology and Environmental Protection, University of Lodz, 90-236 Łódź, Poland

**Keywords:** thiazole derivative, copper complexes, drug design, MCF7

## Abstract

Investigating novel, biologically-active coordination compounds that may be useful in the design of breast anticancer, antifungal, and antimicrobial agents is still the main challenge for chemists. In order to get closer to solving this problem, three new copper coordination compounds containing thiazole-based derivatives were synthesized. The structures of the synthesized compounds and their physicochemical characterization were evaluated based on elemental analysis, ^1^H and ^l3^C nuclear magnetic resonance (NMR), flame atomic absorption spectroscopy (F-AAS), single-crystal X-ray diffraction, thermogravimetric analysis (TGA), and Fourier-transform infrared spectroscopy (FTIR). The pharmacokinetics were studied using SwissADME. The results obtained from the computational studies supported the results obtained from the MTT analysis, and the antimicrobial activity was expressed as the minimum inhibitory concentration (MIC).

## 1. Introduction

New drug design is the perspective process from both aspects of discovery and synthesis. The main scope of the design of new compounds or complexes is based on the knowledge of the biological target. Over the past ten years, attention toward antifungal, antimicrobial, and anticancer properties have risen. Cancer ranks as a leading cause of death in every country of the world. Moreover, based on the statistics from the World Health Organization (WHO), there were an estimated 19.3 million new cancer cases in 2020. Among these, 2.26 million (11.7%) cases refer to female breast cancer [1].

A significant increase in disease cases is mainly associated with fungal infections. Invasive candidiasis is the most common fungal disease among hospitalized patients in developed countries [2,3]. Besides interest in antifungal activity, the development of new antimicrobial agents with novel structures remains the primary goal for finding a solution to increasing bacterial resistance [4,5,6,7]. *Escherichia coli* and *Staphylococcus aureus* are the two main causes of healthcare-associated and hospital-acquired invasive infections in adults [8]. The majority of *E. coli* and *S. aureus* bacteremia occur in individuals aged 65 years and older, with the highest rates in those aged 85 years and older [9,10].

A potentially commercial drug is mostly represented as the organic ligand that inhibits or activates the properties of a biomolecule. The affinity of organic ligands to coordinate transition metal ions has increased in the research field of coordination chemistry. Therefore, this study is focused on thiazole-based derivatives. Nitrogen-containing heterocycles with a sulfur nucleus are an important class of compounds [11,12]. Recent clinical trials have increased the significance of thiazole derivatives due to their application as anti-inflammatory, anticancer, antifungal, antihypertensive, anticonvulsant, and antidiabetic compounds [13,14,15]. The selection of the appropriate metal salts is the second essential step in the architecture of coordination complexes. Metal-containing compounds are broadly used in a wide range of biological systems. The class of complexes on the basis of platinum derivatives is considered to be one of the most effective in the struggle against cancer tumors today, and cisplatin is the most widespread of them in clinical practice. However, its poor solubility in water and high general toxicity [16] serve as the reason for the development of new antitumoral medications on the basis of other metals compounds. Copper has attracted our scientific attention due to its essential role as the structural component of several enzymes and proteins: laccase, ascorbate oxidase, ceruloplasmin, amine oxidases, superoxide dismutase, and cytochrome C oxidase [17,18,19,20]. Endogenous metals are believed to be less toxic to healthy cells; therefore, the antitumor activity of a number of copper-containing coordination compounds has been tested and encouraging results have been obtained [21].

This work is aimed at broadening research on the inhibition and binding affinity of thiazole-based derivatives complexes. To achieve our goal, three organic ligands, 1-[2-thiazol-4-yl-(4-substitutedphenyl)]-4-*n*-propylpiperazines, and three corresponding Cu(II) complexes were investigated. Moreover, the standard procedures for the characterization of synthesized compounds (F-AAS, FTIR, NMR, and TGA), ADME analysis, MTT assay, and microbiology tests were performed.

## 2. Results and Discussion

### 2.1. Crystal Structure Analysis

The crystal structure was determined for ligand L2 (Figure 1). In this case, a good-quality crystal structure (suitable for X-ray measurements) was obtained. The thiazole and phenyl rings are almost co-planar. The dihedral angle between their planes is 4.3 Å. The bond lengths in the thiazole ring are typical for its 2,4-substituted derivatives (Table 1) [22].

The piperazine possesses a chair conformation with the thiazole ring and propyl group bonded in the equatorial positions (trans configuration). All CN bonds of the piperazine ring have similar lengths. The piperazine CNC angle from the thiazole side is significantly larger than that from the propyl side (Table 1). It is a consequence of the partially aromatic character of the C9-N12 bond, which affects the nitrogen hybridization. Its hybridization is between SP3 and SP2 and that enlarges the CNC angle. In turn, the N15-C19 bond is a classical single bond, therefore the CNC angle from the propyl side is a typical tetrahedral angle.

The L2 molecules do not contain the donors of the classical hydrogen bonds. Consequently, the supramolecular structure of L2 is stabilized via non-classical hydrogen bonds, CH•••N, CH•••S, and CH•••Cl, as well as CH•••π interactions. Furthermore, the molecules do not form π-π stacking interactions despite possessing aromatic rings.

The Hirshfeld surface and the 2D fingerprint plot reflect the above-mentioned intermolecular interactions that exist in the structure of L2 (Figure 2). The domination of dispersive interactions is revealed in large contribution of H•••H and H•••C contacts in comparison to the rest (Table 2).

### 2.2. FTIR Spectra

Figure 3 represents the FTIR spectra of pure ligands and synthesized coordination compounds. The spectra of pure ligands do not have any peaks in the region of 3400−3300 cm^−1^. The opposite situation occurs with the spectra of all copper(II) complexes, which contain peaks in this region. Peaks are assigned to ν(NH) vibrations and correspond to the amino group formed during the complexation process due to hydrogen movement [23].

All obtained spectra include peaks in the range of 2700−3000 cm^−1^, which correspond to the C-H stretching vibrations. In the case of pure ligands, the strong absorption peaks of ν(C=N) in the region of 1600−1620 cm^−1^ and 1200−1230 cm^−1^ are observed. After the complexation processes, this vibration mode in the region of 1600−1620 cm^−1^ is shifted to the higher wavenumbers. They are at 1627, 1620, and 1609 cm^−1^ for free ligands L1, L2, and L3, respectively. In the complexes, these bands appear at 1599 cm^−1^ for Cu(L1)_2_Cl_2_, at 1616 cm^−1^ for Cu(L2)Cl_2_, and at 1606 cm^−1^ in the case of Cu(L3)Cl_2_. These modes confirm the involvement of nitrogen atoms in the complexation processes. The absorption peaks of ν(C-S) are observed in the range of 910−925 cm^−1^. They are at 920, 923, and 919 cm^−1^ for free L1, L2, and L3, respectively. Due to the coordination process, the modes of complexes are shifted to 966 cm^−1^ (Cu(L1)_2_Cl_2_), 967 cm^−1^ (Cu(L2)Cl_2_), and 964 cm^−1^ (Cu(L3)Cl_2_) due to the coordination process.

Uncoordinated ligands include modes of β(CH) and γ(CH) vibrations between 1250−1030 and 840−690 cm^−1^. For obtained complexes, they appear in similar ranges.

Thus, it is clear that all pure ligands act as N-donor and S-donor ligands. In the case of L2 and L3, the coordination appears through the sulfur atom from the thiazole ring and the nitrogen atom from the piperazine ring [24]. L2 and L3 ligands act as bidentate. From the FTIR spectra of the L1 ligand and complex based on it, it is clear that nitrogen and sulfur are also involved in the coordination process.

### 2.3. Thermogravimetric Study

Thermogravimetric analysis (TGA) is an alternative, fast, and useful technique for determining the composition of coordination compounds. Therefore, the thermogravimetric curves of three ligands and complexes are presented in Figure 4.

Table 3 demonstrates the decomposition stages of the three pure ligands and complexes with the identified temperature ranges and mass losses.

Based on the obtained results, the thermolysis of all organic ligands is a two-stage process, which starts between 160−190 °C. The most stable is ligand L3 because it decomposes at 190 °C. The first and the biggest mass losses on the TG curves correspond to the destruction of organic ligands. In the range of 400−660 °C, the burning of organic fragments takes place. All organic ligands correspond to the thermal decomposition stability.

The results demonstrate the thermal decomposition of three copper(II) complexes. The most stable is compound Cu(L1)_2_Cl_2_; it starts to decompose at 70 °C. The other coordination compounds are stable up to 50 °C. The first small mass losses overlapping each other correspond to the initial fragmentation and degradation of the organic ligand. Above 175 °C in the case of Cu(L1)_2_Cl_2_, and 250 °C in the case of Cu(L2)Cl_2_ and Cu(L3)Cl_2_, the biggest mass losses on TG curves are observed; they are associated with the further degradation and decomposition of organic ligands. When the temperature rises, decomposition of the residual part of the ligand, loss of chloride anions, and formation of solid-state metal oxide take place. Above 700 °C, pure CuO appears. Higher than expected final mass losses indicate the formation of volatile polymeric Cu_x_Cl_x_ [25].

### 2.4. Biological Activity Predictions

The activity of the synthesized ligands was investigated. The obtained results are shown in Table 4. According to the results, all three ligands have demonstrated a wide range of biological activity.

### 2.5. ADME Analysis

An ADME analysis is used to investigate the properties and calculate the pharmacokinetic parameters of novel compounds. Drug-likeness descriptors selected using the Lipinski and Veber rules were calculated with SwissADME. The Lipinski’s rule of five by notes that good absorption is more likely when the molecular weight (MW) < 500 Da, number of hydrogen bond donors (HBDs) < 5—as shown in molecular docking studies—LogP < 5, and number of hydrogen bond acceptors (HBAs) < 10. Veber rules identified two other relevant descriptors: the number of rotatable bonds (NBR) < 10 and polar surface area (PSA) < 140 Å^2^ [26].

The calculations and analysis of the newly-synthesized organic ligands and coordination compounds indicated that all compounds except Cu(L1)_2_Cl_2_ have no violations of these rules (Figure 5). The red-colored zone corresponds to the oral bioavailability [27].

Additionally, the ADME analysis confirmed the effectiveness of the tested compounds crossing the blood-brain barrier, except for Cu(L1)_2_Cl_2_ (Figure 6) [28].

A ProTox II toolkit was used to classify the ligands and complexes into toxicity classes, which are shown in Table 5.

### 2.6. Antimicrobial Activity

Assays were performed using reference strains of bacteria: *Staphylococcus aureus* ATCC 33591, *Escherichia coli* ATCC 25922, and fungus *Candida glabrata* ATCC 15126 (Table 6). The tested compounds were dissolved in DMSO solution, which was recommended because of their chemical structures. This solvent has proven antimicrobial activity and the control with a serial dilution of DMSO was also included in the tests [29]. Moreover, nystatin, a conventional antifungal antibiotic, was examined and used as reference material (MIC = 4 µg/co). The activity of synthesized complexes was considered only if the DMSO impact was excluded. Bacterial strains were not sensitive to the tested substances and the MIC values were mainly > 1024 µg/mL, except L1 (128 µg/mL). More promising results were obtained for the *C. glabrata* strain where the MIC values were in the range of 32–128 µg/mL, except for Cu(L1)_2_Cl_2_, where MIC > 1024 µg/mL. The lowest value was obtained for L1 (32 µg/mL). In the cases of L3, Cu(L2)Cl_2_, and Cu(L3)Cl_2_, the MIC was equal to 64 µg/mL. The obtained results from the minimal inhibitory concentration assays indicate that the tested complexes have antifungal activity, but more work in this field is needed to increase and examine their potential.

### 2.7. MTT Analysis

Cell viability was determined through the MTT assay in L929 and MCF7 cells for all compounds (Figure 7). Only two copper complexes, i.e., Cu(L1)_2_Cl_2_ and Cu(L3)Cl_2_, presented cytotoxic effects. The other compounds, i.e., compounds without copper and Cu(L2)Cl_2_, were not cytotoxic in the concentration used. The IC50 values were calculated for Cu(L1)_2_Cl_2_ and Cu(L3)Cl_2_. A differential effect of copper compounds Cu(L1)_2_Cl_2_ and Cu(L3)Cl_2_ were demonstrated in two cell types. The MCF7 cells were more sensitive to both compounds compared to L929 cells. The calculated IC50 value (that is, the concentration of compound which exhibited 50% cell viability) for MCF7 cells was much lower (105.6 µM and 82.64 µM, respectively) than for L929 cells (185.56 µM and 185.86 µM). According to the IC50 results (experimental proof of the safety of the compounds), the compounds do not cause high toxicity in the non-tumorigenic cell line, which indicates a certain selectivity towards tumorigenic cells.

## 3. Materials and Methods

### 3.1. Chemistry

All chemicals used for synthesis were purchased from Sigma-Aldrich (Saint Louis, MO, USA) and Alfa Aesar (Haverhill, MA, USA). All melting points (mp) were measured in open capillaries in an electrothermal apparatus and are uncorrected.

#### 3.1.1. ^1^H, and ^13^C NMR

The ^1^H spectra ligands (L1-L3) were recorded in CDCl_3_ as a solvent in a 600 MHz spectrometer, which is a Bruker Avance III spectrometer at ambient temperature (Appendix A). The chemical shifts are reported in ppm on a scale downfield from tetramethylsilane (TMS) as the internal standard, and the signal patterns are indicated as follows: s = singlet, d = doublet, t = triplet, m = multiplet, br = broad; a number of protons, and J approximates the coupling constant in Hertz. The ^13^C NMR spectra were recorded on a Bruker Avance III in a 600 MHz spectrometer (Appendix A). The TLC data were obtained with Merck silica gel 60F254 aluminum sheets.

#### 3.1.2. Flame Atomic Absorption Spectroscopy

The content of Cu(II) in the solid complex was determined by the F-AAS spectrometer with a continuum source of light and using air/acetylene flame (Analytik Jena, contraAA 300, Jena, Germany). The absorbance was measured at analytical spectral line 324.7 nm. The limit of quantification was 0.04 mg/L. The solid sample was decomposed using the Anton Paar Multiwave 3000 closed system instrument. Mineralization was carried out for 50 min at 240 °C under a pressure of 60 bar. The contents of carbon, hydrogen, and nitrogen were determined by a Vario-micro company Elementar Analysensysteme GmbH.

#### 3.1.3. Fourier-Transform Infrared Spectroscopy

FTIR spectra were recorded with an IRTracer-100 Shimadzu Spectrometer (4000–600 cm^−1^) with the accuracy of recording set as 1 cm^−1^ using KBr pellets.

#### 3.1.4. Thermogravimetric Analysis

The thermolysis of compounds in the air atmosphere was studied by TG-DTG techniques in the range of 25–800 °C at a heating rate of 10 °C min^−1^; TG and DTG curves were recorded on Netzsch TG 209 apparatus under air atmosphere v = 20 mL min^−1^ using ceramic crucibles—which were used as a reference material.

#### 3.1.5. X-ray Measurement and Crystal Structure Determination

X-ray diffraction data were collected for ligand L2 on an XtaLAB Synergy Dualflex Pilatus 300K diffractometer (Rigaku Corporation, Tokyo, Japan). Using Olex2 [30], the structure was solved with the SHELXT [31] using Intrinsic Phasing and refined with the SHELXL [32] using Least Squares minimization. All non-hydrogen atoms were refined anisotropically. All hydrogen atoms were found from the Fourier difference map and refined using the “riding” model. The details concerning X-ray diffraction data and structure refinement are given in Table 7.

### 3.2. Synthesis of Organic Ligands (L1-L3)

The 1-(4-*n*-propyl)piperazine thioamide (1) was directly obtained by the reaction of the 1-*n*-propylpiperazine dihydrobromide with potassium thiocyanate in an aqueous solution [33]. Synthesis of 1-[2-thiazol-4-yl-(4-substitutedphenyl)]-4-*n*-propylpiperazines, outlined in Figure 1, was accomplished according to the procedure described previously in [34].

To a stirred solution of the corresponding 2-bromo-1-phenylethanone (1.60 mmol) in *n*-propanol (16 mL), the 1-(4-*n*-propyl)piperazine thioamide (1) (1.60 mmol) was added. The reaction mixture was heated at 90 °C for 5 h. After cooling, the precipitate was filtered and washed with a small amount of cold *n*-propanol and ethyl ether. The hydrobromide salt/product of the 1-[2-thiazol-4-yl-(4-substitutedphenyl)]-4-*n*-propylpiperazine was mixed with 5% sodium hydroxide aqueous solution for 1 h at room temperature, and then the aqueous layer was extracted with ethyl ether (3 × 30 mL). The organic extracts were washed with water (3 × 30 mL), dried over Na_2_SO_4_, and filtered. The solvent was evaporated to give the corresponding ligand.

L1 (C_16_H_21_N_3_S; M = 287.42); mp 50–51 °C, yield: 0.245g, 53% white solid. ^1^H NMR (600 MHz, CDCl_3_) δ 7.85–7.81 (m, 2H, H_arom_), 7.38–7.34 (m, 2H, H_arom_), 7.28–7.24 (m, 1H, H_arom_), 6.76 (s, 1H, H_thiazole_), 3.61–3.53 (m, 4H, CH_2piper_), 2.61–2.53 (m, 4H, CH_2piper_), 2.39–2.33 (m, 2H, NCH_2_CH_2_CH_3_), 1.59–1.49 (m, 2H, NCH_2_CH_2_CH_3_), 0.93 (t, J = 7.4 Hz, 3H, NCH_2_CH_2_CH_3_).^13^C NMR (151 MHz, CDCl_3_) δ 171.1, 152.0, 135.2, 128.6 (2C), 127.7, 126.2 (2C), 101.5, 60.7, 52.6 (2C), 48.5 (2C), 20.1, 12.0. TLC (dichloromethane:methanol 9:1), Rƒ = 0.76;

L2 (C_16_H_20_ClN_3_S; M = 321.87); mp 102–103 °C, yield: 0.338g, 65% white solid. ^1^H NMR (600 MHz, CDCl_3_), δ 7.82–7.72 (m, 2H, H_arom_), 7.39–7.27 (m, 2H, H_arom_), 6.74 (s, 1H_thiazole_), 3.63–3.49 (m, 4H, CH_2piper_), 2.63–2.52 (m, 4H, CH_2piper_), 2.39–2.31 (m, 2H, NCH_2_CH_2_CH_3_), 1.61–1.49 (m, 2H, NCH_2_CH_2_CH_3_), 0.93 (t, J = 7.4 Hz, 3H, NCH_2_CH_2_CH_3_). ^13^C NMR (151 MHz, CDCl_3_) δ 171.1, 150.7, 133.6, 133.2, 128.6 (2C), 127.3 (2C), 101.7, 60.6, 52.4 (2C), 48.4 (2C), 20.0, 11.9. TLC (dichloromethane:methanol 9:1), Rƒ = 0.72;

L3 (C_17_H_23_N_3_OS; M = 317.45); mp 65–66 °C, yield: 0.421g, 83% white solid. ^1^H NMR (600 MHz, CDCl_3_) δ 7.83–7.70 (m, 2H, H_arom_), 6.96–6.85 (m, 2H, H_arom_), 6.62 (s, 1H_thiazole_), 3.82 (s, 3H,OCH_3_), 3.61–3.50 (m, 4H, CH_2piper_), 2.62–2.50 (m, 4H, CH_2piper_), 2.37–2.33 (m, 2H, NCH_2_CH_2_CH_3_), 1.59–1.47 (m, 2H, NCH_2_CH_2_CH_3_), 0.93 (t, J = 7.4 Hz, 3H, NCH_2_CH_2_CH_3_).^13^C NMR (151 MHz, CDCl_3_) δ 171.1, 159.3, 151.8, 128.3, 127.4 (2C), 113.9 (2C), 99.7, 60.8, 55.4, 52.6 (2C), 48.5 (2C), 20.1, 12.1. TLC (dichloromethane:methanol 9:1), Rƒ = 0.79.

### 3.3. Synthesis of Coordination Compounds

Figure 8 illustrates the general synthesis steps for all complexes.

*Cu(L1)_2_Cl_2_.* The complex was synthesized by mixing the organic ligand, L1 (167.9 mg), and anhydrous CuCl_2_ (81.2 mg), which were dissolved in *n*-propanol (30 mL). The mixture was rapidly stirred and heated for 1.5 h at 40 °C. The formed solid product was filtered and washed with cold *n*-propanol and left for drying.

*Cu(L1)_2_Cl_2_* (C_32_H_42_S_2_N_6_CuCl_2_; M = 709.29); mp 46.7 °C; yield: 15% brown solid. Anal.calcd. for C 54.18%; H 5.96%; N 11.85%; Cu 8.96%; S 9.04%. Found: C 54.31%; H 5.79%; N 11.91% Cu 9.34%; S 9.77%. FTIR spectra (KBr, cm^−1^): ν(NH) 3447,3360; ν(CH) 2966–2607; ν(C=N) 1599; 1230; ν(C=C) 1520; β(CH) 1442; ν(CS) 966, 837.

*Cu(L2)Cl_2_.* The complex was synthesized by mixing the organic ligand, L2 (82.6 mg), and anhydrous CuCl_2_ (36.6 mg), which were dissolved in ethanol (30 mL). The mixture was rapidly stirred and heated for 1.5 h at 40 °C. The formed solid product was filtered and washed with cold ethanol and left for drying.

*Cu(L2)Cl_2_* (C_16_H_20_N_3_SCuCl_3_; M = 456.33); mp 98.5 °C; yield: 40% brown solid. Anal. calcd. for C 42.11%; H 4.41%; N 9.21%; Cu 13.93%; S 7.03%. Found: C 42.99%; H 4.19%; N 8.91%; Cu 13,36%; S 7.87%. FTIR spectra (KBr, cm^−1^): ν(NH) 3446,3355; ν(CH) 2968–2738; ν(C=N) 1616, 1233; ν(C=C) 1527; β(CH) 1276, 1229; ν(CS) 967,835; ν(CCl) 732.

*Cu(L3)Cl_2_.* Complex Cu(L3)Cl_2_ was synthesized by mixing the organic ligand, L3 (80 mg), and anhydrous CuCl_2_ (39.2 mg), which were previously dissolved in ethanol (30 mL). The mixture was rapidly stirred and heated for 1.5 h at 40 °C. The formed solid product was filtered and washed with cold ethanol and left for drying.

*Cu(L3)Cl_2_* (C_17_H_23_SN_3_OCuCl_2_; M = 451.90); mp 56.5 °C; yield: 84% brown solid. Anal.calcd. for C 45.18%; H 5.13%; N 9.13%; Cu 14.06%; S 7.10%. Found: C 44.76%; H 4.99%; N 8.89%; Cu 14.89%; S 6.72%. FTIR spectra (KBr, cm^−1^): ν(NH) 3446, 3356; ν(CH) 2965–2750; ν(C=N) 1606; ν(C=C) 1526; β(CH) 1457; ν(C-O) 1250; ν(CS) 964, 837.

### 3.4. Biological Activity Predictions

The activity prediction of three organic ligands was investigated using the toolkit service http://www.way2drug.com/ (accessed on 25 December 2021) [35]. The Way2Drug portal has been developed and supported by the multidisciplinary team of researchers working in bioinformatics, cheminformatics, and computer-aided drug discovery for about thirty years. The current version predicts several thousand different biological activities based on the structural formula of a drug-like organic compound.

### 3.5. ADME Analysis

The ADME analysis was performed using the Swiss ADME toolkit service. The SwissADME web tool gives free access to a pool of fast, yet robust predictive models for physicochemical properties, pharmacokinetics, drug-likeness, and medicinal chemistry friendliness, among which include in-house proficient methods such as the BOILED-Egg, iLOGP, and Bioavailability Radar [36]. The ProTOX II service was used for the prediction of toxicity for the compounds [37,38,39].

### 3.6. Microbiology

The antimicrobial activity, which was expressed as the minimum inhibitory concentration (MIC) of the synthesized compounds, was tested using the broth microdilution method according to the protocol established by the Clinical and Laboratory Standards Institute [40,41]. The stock solution of the compounds was prepared by dissolving them in DMSO. Serial dilution in the range of 1024–2 µg/mL was made using 96-well plates. Secondly, the bacterial or fungal inoculums were added to the wells with Mueller Hinton II Broth (for bacteria) or RPMI 1640 (for fungus) and diluted substances. Plates were incubated for 24 h and 48 h at 37 °C (bacteria) and 24 h at 37 °C (fungus). Assays were conducted in triplicate. The MIC was determined at the lowest concentration at which the visible growth of microorganisms was inhibited.

### 3.7. Cell Viability Analysis by MTT Assay

Breast cancer cells MCF7 and mouse fibroblast cell line L929 were used as a cell model for testing the effect of compounds on the viability of different types of cells. MCF7 and L929 cells were selected as cancerous and normal cell lines, respectively. Cells were cultured in Dulbecco’s Modified Eagle Medium (DMEM) medium in 96-well plates with a concentration of 10,000 cells per well and incubated for 24 h in a 5% CO_2_ atmosphere at 37 °C. After that, cells were treated with different concentrations—1, 10, 50, 100, and 200 µM—of compounds for 24 h. The viability of cells treated with the tested compounds was assessed by measuring the ability of live cells to metabolize 3-(4,5-dimethylthiazolo-2-yl)-2,5-diphenyl tetrazolium bromide (MTT) to formazan, using a standard protocol. Briefly, after incubation with compounds, the medium of each well was discarded, cells were washed with PBS, and then 100 μL of fresh media with 10 μL MTT solution (5 mg mL^−1^ in PBS) was added to each well. After 4 h incubation at 37 °C, the produced formazan was solubilized by the addition of DMSO and the absorbance of each well was determined at 570 nm using an ELISA reader. The results were expressed as the mean of four replicates as a percentage of control (taken as 100%).

## 4. Conclusions

Numerous articles report that thiazole-based derivatives exhibit many notable pharmacological activities, including antiparasitic, anti-inflammatory, and antitumor activity [42,43,44,45]. This current study has proven the pharmacological activity of three synthesized 1-[2-thiazol-4-yl-(4-substitutedphenyl)]-4-*n*-propylpiperazines. Moreover, three newly-synthesized corresponding Cu(II) complexes were synthesized as well, and were shown to have promising biological activity. Among all synthesized compounds, two copper complexes, Cu(L1)_2_Cl_2_ and Cu(L3)Cl_2_, presented cytotoxic effects. The complexation process has enhanced the pharmacological activity, as the other compounds without copper and Cu(L2)Cl_2_ were not cytotoxic in the used concentration. In conclusion, the tests results from minimal inhibitory concentration assays indicate that the tested complexes have antifungal activity. The preliminary results contained in this paper indicate that the coordination compounds with these ligands are biologically active. Our results can be improved by further research.

## Data Availability

The data presented in this study are available in the article and in the Appendix A.

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
