# Peer review of "Synthesis and Biological Evaluation of Thiazole-Based Derivatives with Potential against Breast Cancer and Antimicrobial Agents"

_ijms, 2022, doi:10.3390/ijms23179844_

Round 1
Reviewer 1 Report
Comments:
The article: “Synthesis and Biological Evaluation of Thiazole-Based Derivatives as Potential Anticancer and Antimicrobial agents” is an interesting article. The authors carried out the synthesis of ligands and Cu complexes. In addition, the authors confirmed the molecular structure of a ligand by single-crystal ray diffraction. The authors also presented the in vitro anticancer activity against breast cancer and in vitro antimicrobial activity against, staphylococcus aureus, Escherichia coli and candida glabrata. Importantly, results reveal that the copper complexes have good results in vitro against candida glabrata.
As we all know, Cancer is an important leading cause of death globally and breast cancer is one of the most diagnosed types of cancer. What is more, the infectious diseases caused by bacteria and fungi nowadays are responsible for the current high mortality rate. Copper complexes are an alternative indeed, since these already exhibited anticancer activity and antimicrobial activity.
Therefore, and as a result of all this, the topic is interesting. However, there are some questions that must be addressed in order to increase the importance of the article.
1- For enhancing the article, the introduction must be reformulated including some points.
A- Including updated statistics data about cancer. Specifically, for breast cancer. The same regarding staphylococcus aureus, Escherichia coli and candida glabrata infectious diseases.
B- Including why the metal complexes are promisor as anticancer and antimicrobial agents, and why copper is an alternative. In this discussion, the author could include the current limitations of cisplatin in cancer treatments, and the antibiotic-resistant bacteria and antibiotic-resistant fungi harm when treating bacteria and fungi infectious.
C- Including copper complexes reported with cancer and antimicrobial activity.
2- Result and discussion
A- The authors must revise the elemental analysis of the Cu(L1)2Cl2. The deviation between calcd. values and the proposed formula is high.
B- The in silico studies are interesting. However, regarding the toxicity, the author could use the cytotoxic results in L929 (non-tumorigenic) to emphasize the not toxicity in vitro of the complexes due to the fact that the complexes are not active in L929 cell line.
C- The authors must include the study of the stability of complexes in an aqueous solution.
D- The authors could include a table in the manuscript with activity results for each application: Cancer, antibacterial and antifungal activity.
E- If possible, the author could include other cancer cell lines for anticancer studies.
F- The authors could include some experiments in order to study the possible mechanism of action of complexes, such as the interaction of the copper complexes and the DNA.
3- Materials and Methods.
A- The authors must study the antifungal activity after 24 and 48 h of incubation at 37 °C for Candida glabrata.
4- Conclusion.
The conclusion must be reformulated including the comments above.
Some points:
Figure 9 could be eliminated.
Raw 167. There is a red comma.
Raw 182. Authors could correct the formula Cu(L3)Cl2 and change it to Cu(L3)Cl2.
Raw 230. Use n-propanol as the authors used in the Raw 228 in the manuscript.
Raw 259. Authors could use the mL abbreviation in the manuscript.
Raw 320. Authors could correct the formula CO2 and change it to CO2
Author Response
Dear Reviewer,
Thank you for taking the time to read and correct our work. Below there are our answers to the review comments:
Comment 1:
Introduction
- Including updated statistics data about cancer. Specifically, for breast cancer. The same regarding staphylococcus aureus, Escherichia coli and candida glabrata infectious diseases.
- Including why the metal complexes are promisor as anticancer and antimicrobial agents, and why copper is an alternative. In this discussion, the author could include the current limitations of cisplatin in cancer treatments, and the antibiotic-resistant bacteria and antibiotic-resistant fungi harm when treating bacteria and fungi infectious.
- Including copper complexes reported with cancer and antimicrobial activity.
We would like to thank you for the comments. Introduction was broadened and fulfilled with the respect to your recommendations. Added information was proved by the corresponding references.
Comment 2:
Results and discussion
- The authors must revise the elemental analysis of the Cu(L1)2Cl2. The deviation between calcd. values and the proposed formula is high.
Thank you for your comment. It was a mistake. We have revised the elemental analysis of the Cu(L1)2Cl2.
- The in silico studies are interesting. However, regarding the toxicity, the author could use the cytotoxic results in L929 (non-tumorigenic) to emphasize the not toxicity in vitroof the complexes due to the fact that the complexes are not active in L929 cell line.
Thank you for your comment. PASS Online predicts over 4000 kinds of biological activity, including pharmacological effects, mechanisms of action, toxic and adverse effects, interaction with metabolic enzymes and transporters, influence on gene expression, etc. Nevertheless, in our case this prediction is suitable only for the ligand. Unfortunately, the program does not recognize ligand-metal coordination.
- The authors must include the study of the stability of complexes in an aqueous solution.
We would like to thank you for the remark. The ligands we use for coordination have never been studied biologically before. The preliminary results included in this paper show that the coordination compounds with these ligands are biologically active and the direction of our research is correct. Therefore, in our further research, we will examine the stability of these complexes in various media and the mechanism of connecting with the target site.
- The authors could include a table in the manuscript with activity results for each application: cancer, antibacterial and antifungal activity.
Thank you very much for this comment. The Table was added to the text.
- If possible, the author could include other cancer cell lines for anticancer studies.
Thank you very much for this comment. Our next publication will include studies on more cancer cell lines
- The authors could include some experiments in order to study the possible mechanism of action of complexes, such as the interaction of the copper complexes and the DNA
Thank you very much for this comment. This mechanism will be taken into account in further research.
Comment 3:
Materials and Methods
- The authors must study the antifungal activity after 24 and 48 h of incubation at 37 °C for Candida glabrata
Thank you for your comment. There is a mistake in the manuscript, incubation was carried out at 37℃. The results were read after 24 hours, we routinely leave the plate for 48 hours, but we did not mention it in the description of the method. The results were the same in both cases. We added this information to the text.
Comment 4:
Conclusion
The conclusion must be reformulated including the comments above.
Some points:
Figure 9 could be eliminated.
Raw 167. There is a red comma.
Raw 182. Authors could correct the formula Cu(L3)Cl2 and change it to Cu(L3)Cl2.
Raw 230. Use n-propanol as the authors used in the Raw 228 in the manuscript.
Raw 259. Authors could use the mL abbreviation in the manuscript.
Raw 320. Authors could correct the formula CO2 and change it to CO2
Thank you very much for all comments. Everything has changed in the text.
Kind regards,
Authors

Reviewer 2 Report
The authors synthesized three organic 1-[2-thiazol-4-yl-(4-substitutedphenyl)]-4-n-propylpiperazines and three corresponding Cu(II) complexes and investigated them from the physicochemical point of view and probed their antimicrobial and cytotoxic activity.
The article is unfortunately written with not enough care. In my opinion the English can be improved and also the description of the results.
The article needs a major revision due to the inaccuracies and hastiness in the description of the results I identified which I’ll present on the following:
Page 1 Lines 20-29: The argumentation in Abstract is not enough logical and can be improved. As an example: “Investigating novel biologically active coordination compounds… is still the main challenge for chemists. Therefore, this current research work is based on the synthesis of new copper containing thiazole-based derivatives. The structures of the synthesized compounds and physicochemical characterization were evaluated based on…”
Page 1 and 2 Lines 33- 59: The same awkwardness in expressing the ideas is still present in the Introduction. Some examples: “New drug design is the perspective process from both aspects of searching and synthesis. The main scope of the design of new compounds or complexes is based on the knowledge of the biological target.” And another one: “the development of new antimicrobial agents with novel structures remains the primary goal for the SOLUTION of increasing bacterial resistance” And another one: “A commercially potential drug is mostly represented as the organic ligand that inhibits or activates the properties of a biomolecule.” And another one “The affinity of organic ligands to coordinate transition metal ions has risen the research field of coordination chemistry. Therefore, this study is focused on the thiazole-based derivatives.” And another one: “Recent clinical trials have increased the SYNTHETIC significance of thiazole derivatives” And another one: “A wide range of biological systems IS WORKING in cooperation with the CORRESPONDING metal ions. Among a HUGE variety, COPPER HAS ATTRACTED OUR SCIENTIFIC ATTENTION…” etc. In my opinion the Introduction can/must be improved and the ideas expressed with more care.
Page 2 and 3 Lines 60-84 The same hastiness is present in all Section 2 Results and Discussion. Examples: “The crystal structure was determined for L2 (Figure 1). In this case, a good quality crystal structure (suitable for X-ray measurements) was obtained. The molecule differs in the orientation of the piperazine ring.” The analysis of the crystal structure of L2 compound it is too “telegraphic” without a good argumentation and even if authors are talking about compound L2, Table 1 contains “Selected structural data of L1.”
Page 3 and 4 Lines 85-109 – Subsection2.2 FTIR spectra: The imprecision persist and the infrared vibrational spectra is analyzed again with not enough attention. As an example the authors mention: “In the case of pure ligands, the strong absorption peaks of ν(C=N) in the region 1600−1620 cm^(-1) and 1200−1230 cm^(-1) are observed. After the complexation processes, THIS VIBRATION MODE is shifted to the higher wavenumbers, which can confirm the involvement of nitrogen atoms in the complexation processes.” There are at least TWO vibrational peaks both associated to the stretching C=N, as the authors mention, but they are not described with enough details (what kind of stretching ?) and in Figure 3, these shifts ”to the higher wavenumbers” are not visible and they are not specified in any way. We have only to trust the authors! Another awkward phrase is “In the absorption of the S-donor ligand appears modes of ν(C-S) in the range of 910−925 cm-1, RESPECTIVELY.” And the argumentation is too vague “For obtained complexes, they appear in similar ranges. Thus, this is clear, that…” And why the authors cite the reference [17], Abu-Yamin et al. paper on Lanthanide complexes at the end of the conclusion “In the case of L2 and L3 – the coordination appears through the sulfur atom from the thiazole ring and the nitrogen atom from the piperazine ring [17].” What is the argument ? The authors have to write it down not let the readers to consult the reference [17], if the reference is effectively needed. On the following the authors mention: “From the FTIR spectra of the L1 ligand and complex based on it, it is clear that also nitrogen and sulfur are involved in the coordination process. However, we assume that this ligand act as monodentate, which correlates with the elemental analysis, F-AAS, and thermogravimetric results.” This seems to be only an opinion because is not enough explained why elemental analysis, F-AAS, and thermogravimetric results are more relevant than those obtained in FTIR spectra.
Page 4-6 Lines 110-135 The same haste and “telegraphic” way of writing of the sentences appear in the subsection 2.3 Thermogravimetric study. Some examples: “Thermogravimetric analysis (TGA) is an alternative, fast, and useful technique for the determining of the ACHIEVED composition of coordination compounds. Therefore, thermograms of three ligands Figure 4 and three complexes Figure 5 were investigated. Table 3 demonstrates…”, “Based on the ACHEVED RESULTS thermolysis of all organic ligands is two stage process which starts between 160−190 °C. The most stable is L3. It decomposes at 190 °C…. The OBTAINED RESULTS demonstrate the thermal decomposition of three copper (II) complexes. The most stable is Cu(L1)2Cl2. It starts to decompose at 70 °C.” It would be useful to represent thermograms from Figure 4 in the same graph in this way the values for the three compounds can be compared. The graph will be bigger and I personally think that using different colors a way can be found and the graph will not be too loaded. The same thing also for the Figure 5. Or, at least, thermograms can be stacked one on top of the other.
Page 6 Lines 136-142 Subsection 2.4 Activity predictions. “The ACTIVITY of the synthesized ligands was investigated. THE OBTAINED RESULTS are shown in Table 4. According to THE ACHIEVED RESULTS, all 3 ligands have shown effective activity in neurodegenerative disease treatment – more than 56%.” What kind of “activity” and why you are presenting only neurodegenerative disease and not also about the others where the probability to be active seems to be higher?
Page 7-8 Lines 143-164. ADME Analysis. In Line 149 the authors intended to say “no more than 10” in spite of “number of hydrogen bond acceptors (HBAs) < 0.”
Page 9 at line 198 the authors mention that “The 1H ----,---- and 13C NMR SPECTRA LIGANDS (L1-L3) were recorded in CDCl3 as a solvent in a 600 MHz spectrometer, a Bruker Avance III spectrometer at ambient temperature.” and then at line 204 “The 13C NMR spectra were recorded in a 600 MHz spectrometer, a Bruker Avance III (150 MHz).” The authors probably intended to say that copper complexes were measured with the Bruker spectrometer at 150 MHz. In my opinion it will be better to split the subsection 3.1 - which is not very properly named “Chemistry” - to the corresponding analysis methods, NMR, thin-layer chromatography, F-AAS, FTIR, TGA etc. Only X-ray crystal structure diffraction is presented separately in subsection 3.4.
Page 13 Line 341 Conclusions. I hope that the following conclusion can be reformulated “Among all synthesized compounds, two copper complexes Cu(L1)2Cl2 and Cu(L3)Cl2 ejected cytotoxic effects. The complexation process has enhanced the pharmacological activity, as the other compounds without copper and Cu(L2)Cl2 were not cytotoxic in the used concentration.” “Ejected” sounds strange (and it appears also at line 182) even if is probably a “technical” term and can be probably replaced by “presented”. The description is somehow clumsy. The last sentence in the article is again inelegant “The obtained results indicate the direction of further research.” It can be replaced by “Our results can be improved on the following by further research.” or something similar.
Despite the experimental work done, the results are not presented with enough care. My general impression regarding the article is not very favorable and it was somehow difficult to read. Principally Section 2. Results and discussion must be re-written with more attention and care. Even the Abstract and the Conclusions are not enough clear and contain clumsy phrases.
Author Response
Dear Reviewer,
Thank you for taking the time to read and correct our work. Below there are our answers to the review comments:
Comment 1:
Page 1 Lines 20-29: The argumentation in Abstract is not enough logical and can be improved. As an example: “Investigating novel biologically active coordination compounds… is still the main challenge for chemists. Therefore, this current research work is based on the synthesis of new copper containing thiazole-based derivatives. The structures of the synthesized compounds and physicochemical characterization were evaluated based on…”
We would like to thank you for the remarks. We have changed it with the respect to your recommendations.
Comment 2:
Page 1 and 2 Lines 33- 59: The same awkwardness in expressing the ideas is still present in the Introduction. Some examples: “New drug design is the perspective process from both aspects of searching and synthesis. The main scope of the design of new compounds or complexes is based on the knowledge of the biological target.” And another one: “the development of new antimicrobial agents with novel structures remains the primary goal for the SOLUTION of increasing bacterial resistance” And another one: “A commercially potential drug is mostly represented as the organic ligand that inhibits or activates the properties of a biomolecule.” And another one “The affinity of organic ligands to coordinate transition metal ions has risen the research field of coordination chemistry. Therefore, this study is focused on the thiazole-based derivatives.” And another one: “Recent clinical trials have increased the SYNTHETIC significance of thiazole derivatives” And another one: “A wide range of biological systems IS WORKING in cooperation with the CORRESPONDING metal ions. Among a HUGE variety, COPPER HAS ATTRACTED OUR SCIENTIFIC ATTENTION…” etc. In my opinion the Introduction can/must be improved and the ideas expressed with more care.
We would like to thank you for the remarks. We have changed some sentences due to your recommendations and added new information.
Comment 3:
Page 2 and 3 Lines 60-84 The same hastiness is present in all Section 2 Results and Discussion. Examples: “The crystal structure was determined for L2 (Figure 1). In this case, a good quality crystal structure (suitable for X-ray measurements) was obtained. The molecule differs in the orientation of the piperazine ring.” The analysis of the crystal structure of L2 compound it is too “telegraphic” without a good argumentation and even if authors are talking about compound L2, Table 1 contains “Selected structural data of L1.”
We would like to thank you for the remarks. You were right, there was made a mistake. L1 was changed to L2. Moreover, we have slightly rewritten this part.
Comment 4:
Page 3 and 4 Lines 85-109 – Subsection2.2 FTIR spectra: The imprecision persist and the infrared vibrational spectra is analyzed again with not enough attention. As an example the authors mention: “In the case of pure ligands, the strong absorption peaks of ν(C=N) in the region 1600−1620 cm^(-1) and 1200−1230 cm^(-1) are observed. After the complexation processes, THIS VIBRATION MODE is shifted to the higher wavenumbers, which can confirm the involvement of nitrogen atoms in the complexation processes.” There are at least TWO vibrational peaks both associated to the stretching C=N, as the authors mention, but they are not described with enough details (what kind of stretching ?) and in Figure 3, these shifts ”to the higher wavenumbers” are not visible and they are not specified in any way. We have only to trust the authors! Another awkward phrase is “In the absorption of the S-donor ligand appears modes of ν(C-S) in the range of 910−925 cm-1, RESPECTIVELY.” And the argumentation is too vague “For obtained complexes, they appear in similar ranges. Thus, this is clear, that…” And why the authors cite the reference [17], Abu-Yamin et al. paper on Lanthanide complexes at the end of the conclusion “In the case of L2 and L3 – the coordination appears through the sulfur atom from the thiazole ring and the nitrogen atom from the piperazine ring [17].” What is the argument ? The authors have to write it down not let the readers to consult the reference [17], if the reference is effectively needed. On the following the authors mention: “From the FTIR spectra of the L1 ligand and complex based on it, it is clear that also nitrogen and sulfur are involved in the coordination process. However, we assume that this ligand act as monodentate, which correlates with the elemental analysis, F-AAS, and thermogravimetric results.” This seems to be only an opinion because is not enough explained why elemental analysis, F-AAS, and thermogravimetric results are more relevant than those obtained in FTIR spectra.
We would like to thank you for the remarks. Due to your comments, we clarify the information. This reference 17 (in revised version of the manuscript) was provided as confirmation of the proposed coordination mechanism using an example of a functionally similar organic compound published in MDPI "Pharmaceuticals”. We removed the sentence: However, we assume that this ligand act as monodentate, which correlates with the elemental analysis, F-AAS, and thermogravimetric results.” from the text.
Comment 5:
Page 4-6 Lines 110-135 The same haste and “telegraphic” way of writing of the sentences appear in the subsection 2.3 Thermogravimetric study. Some examples: “Thermogravimetric analysis (TGA) is an alternative, fast, and useful technique for the determining of the ACHIEVED composition of coordination compounds. Therefore, thermograms of three ligands Figure 4 and three complexes Figure 5 were investigated. Table 3 demonstrates…”, “Based on the ACHEVED RESULTS thermolysis of all organic ligands is two stage process which starts between 160−190 °C. The most stable is L3. It decomposes at 190 °C…. The OBTAINED RESULTS demonstrate the thermal decomposition of three copper (II) complexes. The most stable is Cu(L1)2Cl2. It starts to decompose at 70 °C.” It would be useful to represent thermograms from Figure 4 in the same graph in this way the values for the three compounds can be compared. The graph will be bigger and I personally think that using different colors a way can be found and the graph will not be too loaded. The same thing also for the Figure 5. Or, at least, thermograms can be stacked one on top of the other.
We would like to thank you for the remarks. Some of the sentences were rewritten due to your recommendations. Overlapping thermograms can cause defocusing. Therefore, the decision was made to combine the two figures into one. Moreover, Table 3 demonstrates and compares the decomposition stages of the three complexes and ligands with the identified temperature range and mass loss. Thank you for your suggestions.
Comment 6:
Page 6 Lines 136-142 Subsection 2.4 Activity predictions. “The ACTIVITY of the synthesized ligands was investigated. THE OBTAINED RESULTS are shown in Table 4. According to THE ACHIEVED RESULTS, all 3 ligands have shown effective activity in neurodegenerative disease treatment – more than 56%.” What kind of “activity” and why you are presenting only neurodegenerative disease and not also about the others where the probability to be active seems to be higher?
Thank you for the comments. The activity prediction was renamed as “the biological activity predictions”. Due to your recommendations, this part was changed.
Comment 7:
Page 7-8 Lines 143-164. ADME Analysis. In Line 149 the authors intended to say “no more than 10” in spite of “number of hydrogen bond acceptors (HBAs) < 0.”
Thank you for the comments. You were right, there was made a mistake. “number of hydrogen bond acceptors (HBAs) < 10.” This sentence was corrected.
Comment 8:
Page 9 at line 198 the authors mention that “The 1H ----,---- and 13C NMR SPECTRA LIGANDS (L1-L3) were recorded in CDCl3 as a solvent in a 600 MHz spectrometer, a Bruker Avance III spectrometer at ambient temperature.” and then at line 204 “The 13C NMR spectra were recorded in a 600 MHz spectrometer, a Bruker Avance III (150 MHz).” The authors probably intended to say that copper complexes were measured with the Bruker spectrometer at 150 MHz. In my opinion it will be better to split the subsection 3.1 - which is not very properly named “Chemistry” - to the corresponding analysis methods, NMR, thin-layer chromatography, F-AAS, FTIR, TGA etc. Only X-ray crystal structure diffraction is presented separately in subsection 3.4.
Thank you for the comments. We have taken your suggestions into account and provided a separate description of each method.
Comment 9:
Page 13 Line 341 Conclusions. I hope that the following conclusion can be reformulated “Among all synthesized compounds, two copper complexes Cu(L1)2Cl2 and Cu(L3)Cl2 ejected cytotoxic effects. The complexation process has enhanced the pharmacological activity, as the other compounds without copper and Cu(L2)Cl2 were not cytotoxic in the used concentration.” “Ejected” sounds strange (and it appears also at line 182) even if is probably a “technical” term and can be probably replaced by “presented”. The description is somehow clumsy. The last sentence in the article is again inelegant “The obtained results indicate the direction of further research.” It can be replaced by “Our results can be improved on the following by further research.” or something similar.
We would like to thank you for the remarks. We have taken your suggestions into account.
Kind regards,
Authors

Round 2
Reviewer 1 Report
The authors have solved the majority of all the suggestions. However, and in addition to this, the following points need to be taken into account.
1- In line 38 the first sentence seems to be incomplete. It is possible that the authors are referring to the increasing number of deaths in the world caused by fungal infections.
2- In lines 42 and 44, the authors could write the names S. aureus, E. coli in italics.
3- The authors use in silico studies to predict the toxicity of both ligands and metal compounds, as shown in table 5. Toxicity prediction results. However, the authors have obtained a good toxicity result in the L929 cell line (non-tumorigenic). According to the IC50 results, the compounds do not generate high toxicity in the L929 cell line, which indicates a certain selectivity towards tumorigenic cells. These results in themselves are in vitro experimental proof of the safety of the compounds. For this reason, the authors can add a phrase indicating the selectivity, and the non-toxicity of the compounds in non-tumorigenic cells. This is important, because toxicity is a determining factor in the development of new metallodrugs. Furthermore, the compounds may possibly be more active in other cancer cell lines.
Author Response
Dear Reviewer,
Thank You for taking the time to read and correct our work. Below there are our answers to the review comments:
Comment 1:
In line 38 the first sentence seems to be incomplete. It is possible that the authors are referring to the increasing number of deaths in the world caused by fungal infections.
Thanks for Your comment. We have changed this sentence due to the incomplete version.
Comment 2:
In lines 42 and 44, the authors could write the names S. aureus, E. coli in italics.
Thank You for the comment. We have made the changes.
Comment 3:
The authors use in silico studies to predict the toxicity of both ligands and metal compounds, as shown in table 5. Toxicity prediction results. However, the authors have obtained a good toxicity result in the L929 cell line (non-tumorigenic). According to the IC50 results, the compounds do not generate high toxicity in the L929 cell line, which indicates a certain selectivity towards tumorigenic cells. These results in themselves are in vitro experimental proof of the safety of the compounds. For this reason, the authors can add a phrase indicating the selectivity, and the non-toxicity of the compounds in non-tumorigenic cells. This is important, because toxicity is a determining factor in the development of new metallodrugs. Furthermore, the compounds may possibly be more active in other cancer cell lines.
Thank You for the comments. We have taken Your suggestions into account and added the following sentence “According to the IC50 results (experimental proof of the safety of the compounds), the compounds do not cause high toxicity in the non-tumorigenic cell line, which indicates a certain selectivity towards tumorigenic cells.” in 2.7 part.
Kind regards,
Authors
